

# Clinical value of hemoglobin, albumin, lymphocyte, and platelet indexes in predicting lymph node metastasis and recurrence of endometrial cancer: a retrospective study

Ying Xiong, Yuanyuan Yong and Yanhua Wang

Department of Gynecology, General Hospital of Ningxia Medical University, Yinchuan, China

## ABSTRACT

**Objective:** To study the clinical importance of hemoglobin, albumin, lymphocyte, and platelet (HALP) indexes in predicting lymph node metastasis and recurrence of endometrial cancer.

**Methods:** From July 2016 to July 2020, 158 patients suffering from endometrial cancer who visited the gynecology department of General Hospital of Ningxia Medical University from were collected. Employing the X-Tiles program, the ideal HALP cut-off value was established, and the patients were separated into low and high HALP groups. Univariate and multivariate analysis were used to determine the relationship between HALP score and lymph node metastasis and recurrence of endometrial cancer.

**Results:** The optimal cut-off value of HALP score was established to be 22.2 using X-Tiles software, and the patients were separated into high HALP group (HALP score > 22.2, with 43 cases) and low HALP group (HALP score ≤ 22.2, 115 cases). Endometrial cancer patients' HALP scores were strongly connected with differentiation, the degree of myometrial invasion, and lymph node metastasis ($P < 0.05$), although not with age, menopausal status, or stage ($P > 0.05$). Multivariate logistic regression analysis revealed that the HALP score (OR = 2.087) was the influencing factor for lymph node metastasis ($P < 0.05$). The ROC curve suggested that the AUC of HALP score in predicting lymph node metastasis was 0.871, which had high diagnostic value. When compared to patients without recurrence, HALP scores of patients with recurrence were considerably lower ($P < 0.05$). Multivariate logistic regression analysis showed that HALP score (OR = 2.216) was the influencing factor for the occurrence of lymph node metastasis ($P < 0.05$). The ROC curve suggested that the AUC of HALP score in predicting relapse was 0.855, with high diagnostic value.

**Conclusion:** The HALP score shows good predictive performance in predicting lymph node metastasis and recurrence of endometrial cancer, and has high clinical value, which helps in improving the accuracy and effectiveness of clinical diagnosis and prognosis research.

Corresponding author
Ying Xiong,
young19780806@126.com

## INTRODUCTION

Endometrial cancer (EC) represents one of the three main malignant tumors related to the female reproductive system and is highly prevalent in developed nations (*van den Heerik et al., 2021*). The typical early symptoms are irregular vaginal bleeding or fluid drainage, Epidemiological surveys show that 70% to 75% of EC occurs in postmenopausal women, with a high incidence age ranging from 75 to 79 years old (*Terzic et al., 2021*). The main treatment plan is surgery, supplemented by radiotherapy and chemotherapy. The National Comprehensive Cancer Network (NCCN) guidelines emphasize comprehensive staging surgery (systematic pelvic and para-aortic lymph node resection), including patients having early tumors limited to the uterus, in order to identify whether there is lymph node metastasis, provide accurate staging and prognosis prediction, and further guide postoperative adjuvant therapy (*Koh et al., 2018*). Lymph node metastasis is of great significance for the treatment and prognosis of endometrial cancer. Lymph node metastasis usually means that cancer cells have begun to spread to other areas, and the patient's treatment plan may need to be adjusted, such as pelvic lymph node dissection surgery or adjuvant radiotherapy. In addition, lymph node metastasis also affects the prognosis of patients, which may indicate that the tumor has entered a more advanced stage. The prognosis of patients is highly correlated with early diagnosis and therapy. The accuracy of preoperative evaluation is very important for the choice of treatment plan.

Recent studies have identified a new inflammation index called HALP, comprised of hemoglobin, albumin, lymphocytes, and platelets, which has proven to be a good prognostic indicator in gastric, colorectal, renal, and bladder cancers (*Xu et al., 2023*). (1) Hemoglobin is the main molecule that carries and transports oxygen in the body. Hypoxia is an important factor in tumor metabolism, survival, invasion, migration, angiogenesis, and resistance to chemotherapy or radiation therapy. (2) Albumin is the main protein in plasma and a reflection of body nutrition. (3) Lymphocytes are the main functional cells in the immune response of the body. During the formation and growth of tumors, as heterogeneous antigens, it can stimulate the body to produce immune responses and generate a large number of lymphocytes. When a tumor undergoes immune escape, the surface of the tumor cells can express antigens that inhibit immune cells, which bind to immune cells and lead to their apoptosis. (4) Platelets have a vital part in the process of thrombosis and anticoagulation. The specific theoretical basis can be found in the discussion section (*Ekinci et al., 2022*; *Farag et al., 2023*; *Kaya et al., 2021*). This study will retrospectively explore the predictive value of HALP score for lymph node metastasis and recurrence of EC.

## MATERIALS AND METHODS

### General data

This study examined 158 EC patients who consulted General Hospital of Ningxia Medical University's gynecology department between July 2016 and July 2020. They ranged in age from 29 to 81, having an average age of 55.54 ± 8.64 years, with 57 premenopausal and 101 menopausal cases. In 2009, the International Federation of Gynecology and Obstetrics (FIGO) adopted staging standards (*Mariani, Dowdy & Podratz, 2009*; *Guo et al., 2019*). There were 66 cases with high differentiation, 61 cases having moderate differentiation, and 31 cases had low differentiation; 21 cases of lymph node metastasis and 137 cases without lymph node metastasis; 114 cases in stage I, 16 cases in stage II, 26 cases in stage III, and 2 cases in stage IV; The depth of myometrial invasion was <1/2 in 109 cases and ≥1/2 in 49 cases. The whole design of this study was shown in the flowchart in Fig. 1.

Inclusion criteria: (1) EC were confirmed by histological or cytological pathological examination. Exclusion criteria: (1) patients suffering from other malignant tumors; (2) patients having severe systemic infection; (3) Patients having complications related to serious diseases of blood system and endocrine system; (4) Patients who received radiotherapy, chemotherapy, and estrogen therapy before inclusion in the study; (5) Severe heart, liver, and kidney dysfunction, inability to cooperate with treatment, or intolerance during the treatment process. The medical ethics committee at General Hospital of Ningxia Medical University gave its approval for this investigation and agreed to waive the informed consent.

### Methods

#### Collection of clinical data

Clinical data such as age, gender, menopause, differentiation grade, lymph node metastasis, staging, and myometrial invasion were collected by the hospital medical record management system. The XE-5000 blood cell analyzer from Sysmex Company (Kobe, Japan) was used to analyze the blood routine results, while the AU5800 automatic biochemical analyzer from Beckman Coulter Company (Brea, CA, USA) was used to analyze the biochemical results. All patients had their fasting venous blood drawn in the morning a week prior to the start of treatment. An automatic blood analyzer was used to determine the hemoglobin level, lymphocyte, platelet and monocyte count, and automatic biochemical analyzer was used to determine the human blood albumin level.

#### HALP score

One week prior to the start of the first chemotherapy session, hematological measures such as serum albumin, hemoglobin, lymphocyte, and platelet counts were taken. The formula used to determine the HALP score is as follows: HALP score = hemoglobin (g/L) × albumin (g/L) × lymphocyte (/L)/platelet (/L).

### Follow up

After the treatment, patients were followed up by outpatient review, telephone or text message, *etc.* Within 1 year after diagnosis, patients had been followed up every 3 months,

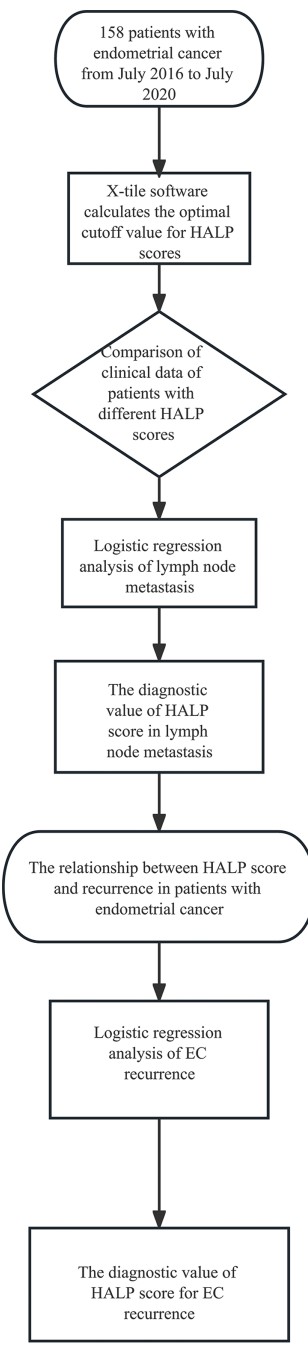

**Figure 1 Flowchart of the study.**

and every 6 months within 3 years. The follow-up included evaluation of patients' general condition, recurrence and progression, and survival outcome through physical examination, laboratory examination, and ultrasound examination. The deadline for follow-up was May 1, 2023.

## Statistical analysis

The experimental data were analyzed employing SPSS21.0. The optimal cutoff value of HALP score was calculated using X-tile 3.6.1 (Yale University, New Haven CT, USA) software. Measurement data that followed a normal distribution were represented by $\bar{X} \pm S$ and the independent sample t-test was utilized for comparing the two groups. Counting data were expressed as examples or rates, and the two groups were compared *via* the $\chi^2$ test. Several groups of graded data were compared employing the Kruskal Wallis rank sum test. Factors with significant differences in single factors were included in multivariate logistic regression analysis. The influencing factors of AMI were examined employing a multivariate logistic regression model, and the diagnostic value was assessed utilizing a receiver operating characteristic (ROC) curve, the optimal critical value is selected according to the Receiver operating characteristic, and the diagnostic value with the largest Youden index is determined. Statistical significance was established at $P < 0.05$.

# RESULTS

## Cut-off value of HALP score

In this study, X-tile software was used for calculating the optimal cut-off value of HALP score as 22.2, with $P = 0.0149$, as shown in Fig. 2. Then, according to HALP score, patients were separated into the high HALP score group (HALP score > 22.2) and low HALP score group (HALP score ≤ 22.2). There were 43 and 115 patients in the two groups, accounting for 27.22% and 72.78%, respectively.

## Relationship between the HALP score and clinical pathology of EC patients

As indicated in Table 1, the HALP score had a strong correlation with differentiation, the degree of myometrial invasion, and lymph node metastasis in EC patients ($P < 0.05$), though not with age, menopausal status, or stage ($P > 0.05$).

## Logistic regression analysis

In a univariate analysis, statistically significant variables (HALP score) were selected as independent variables, and the occurrence of lymph node metastasis was taken as the dependent variable (0 = no, 1 = yes), a multivariate logistic regression analysis was conducted. The results showed that HALP score (OR = 2.087) was the influencing factor for the occurrence of lymph node metastasis ($P < 0.05$), as indicated in Table 2.

## Diagnostic value of HALP score for lymph node metastasis

HALP score was used as the independent variable and lymph node metastasis as the dependent variable (1 = yes, 0 = no), then ROC curve was plotted. The results indicated that the AUC of HALP score in predicting lymph node metastasis was 0.871, which had high diagnostic value, as demonstrated in both Table 3 and Fig. 3.
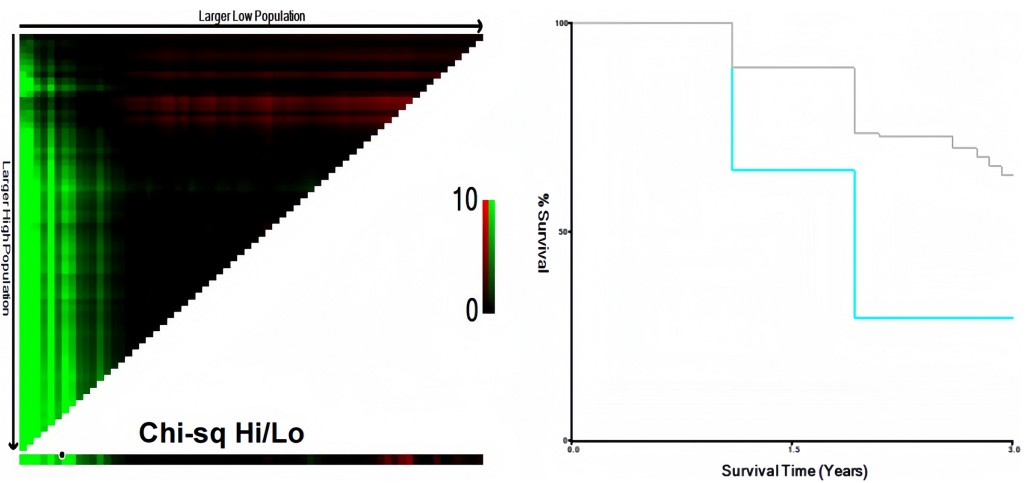

Figure 2 Cut-off value for the HALP score was determined using the X-tile software.

Table 1 Relationship between HALP score and clinical pathology of EC patients.

| Clinical data | n | HALP score | | Statistic | P |
|---|---|---|---|---|---|
| | | >22.2 (n = 43) | ≤22.2 (n = 115) | | |
| Age | | | | | |
| <60 years | 98 | 26 | 72 | 0.247 | 0.805 |
| ≥60 years | 60 | 17 | 43 | | |
| Menopause | | | | | |
| No | 57 | 20 | 37 | 1.670 | 0.095 |
| Yes | 101 | 23 | 78 | | |
| Stage | | | | | |
| Stage I | 114 | 31 | 83 | 1.827 | 0.609 |
| Stage II | 16 | 6 | 10 | | |
| Stage III | 26 | 6 | 20 | | |
| Stage IV | 2 | 0 | 2 | | |
| Differentiation | | | | | |
| High | 66 | 28 | 38 | 15.372 | 0.001 |
| Moderate | 61 | 7 | 54 | | |
| Low | 31 | 8 | 23 | | |
| Depth of myometrial invasion | | | | | |
| <1/2 | 109 | 36 | 73 | 2.448 | 0.014 |
| ≥1/2 | 49 | 7 | 42 | | |
| Lymph node metastasis | | | | | |
| No | 137 | 42 | 95 | 2.483 | 0.013 |
| Yes | 21 | 1 | 20 | | |

**Table 2 Logistic regression model analysis.**

| Hazard | β | SE | Wardχ2 | P | OR [95% CI] |
|---|---|---|---|---|---|
| HALP score | 0.665 | 0.323 | 4.743 | 0.000 | 2.087 [1.569~3.691] |

**Table 3 The diagnostic value of HALP score for lymph node metastasis.**

| Index | AUC | [95% CI] | Optimal cut-off value | Specificity | Sensitivity |
|---|---|---|---|---|---|
| HALP score | 0.871 | [0.754~0.976] | 24.26 | 0.812 | 0.765 |

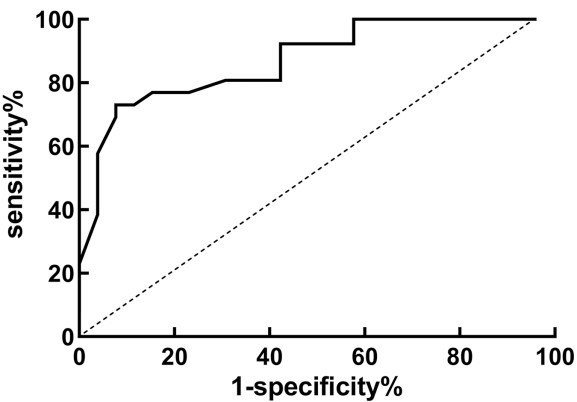

**Figure 3 ROC curve of HALP score for diagnosis of lymph node metastasis in EC patients.**

## The relationship between HALP score and recurrence in EC patients

The HALP scores of EC patients having recurrence were substantially lower than the patients without recurrence ($P < 0.05$), as demonstrated in Table 4.

## Logistic regression analysis

Univariate analysis was performed, independent variables were defined as those that were statistically significant, and recurrence of EC patients was taken as the dependent variable (0 = no, 1 = yes), and multivariate Logistic regression analysis was performed. The results showed that the HALP score (OR = 2.216) was an influencing factor for recurrence ($P < 0.05$), as demonstrated in Table 5.

## Diagnostic value of HALP score for recurrence in EC patients

HALP score was used as the test variable and lymph node metastasis as the dependent variable (1 = yes, 0 = no), the ROC curve was plotted. Results indicated that the AUC of HALP score for predicting recurrence was 0.855, which had high diagnostic value, as depicted in both Table 6 and Fig. 4.

**Table 4 The relationship between HALP score and recurrence in EC patients.**

| Clinical data | n | HALP score | |
|---|---|---|---|
| | | >22.2 (n = 43) | ≤22.2 (n = 115) |
| Recurrence | | | |
| No | 39 | 18 | 21 |
| Yes | 119 | 25 | 94 |
| Statistic | | 3.062 | |
| P | | 0.002 | |

**Table 5 Logistic regression model analysis.**

| Hazard | β | SE | Wardχ2 | P | OR [95% CI] |
|---|---|---|---|---|---|
| HALP score | 0.796 | 0.365 | 4.752 | 0.000 | 2.216 [1.084~4.532] |

**Table 6 Diagnostic value of HALP score for recurrence in EC patients.**

| Index | AUC | [95% CI] | Optimal cut-off value | Specificity | Sensitivity |
|---|---|---|---|---|---|
| HALP score | 0.855 | [0.752~0.958] | 21.15 | 0.809 | 0.754 |

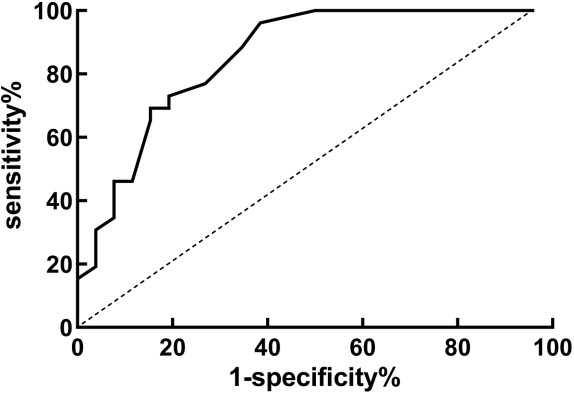

**Figure 4 ROC curve of HALP score for diagnosis of recurrence in EC patients.**

## DISCUSSION

Malignant cells exhibit traits of proliferation, invasion, and metastasis, and the systemic inflammatory immunological milieu can contribute to these characteristics.

The nutritional and metabolic status of the host plays a role as well. The HALP score is a composite measure of hemoglobin, lymphocyte, platelet count, and albumin that reflects both inflammation and nutritional status (*Wang et al., 2023*; *Duran et al., 2022*).
We examined its use in predicting lymph node metastasis and recurrence in EC patients in this study.

Systemic inflammation stimulates angiogenesis, immunosuppression and supports the formation of microenvironment, thus promoting the initiation, progression and metastasis of tumor cells (*Calderillo Ruiz et al., 2022*). Recently, research has focused on the impact of nutritious status on a cancer patient's prognosis. Advanced tumors in particular are a chronic wasting illness. In recent years, several investigations established that systemic inflammation and nutritional status are connected to cancer patient's prognosis (*Rock et al., 2012*). HALP score, composed of lymphocytes, albumin, hemoglobin and platelets, has recently been employed for predicting the prognosis of patients suffering from different forms of cancers (*Guo et al., 2019*; *Xu et al., 2020*; *Peng et al., 2018*; *Yang et al., 2020*; *Fang et al., 2023*). The data of 82 prostate cancer patients were retrospectively analyzed by *Guo et al. (2019)* and the findings revealed that HALP was an independent prognostic factor for prostate cancer patients who had multiple or limited metastases. A study of 582 pancreatic cancer patients revealed that preoperative HALP score was strongly associated to clinical features, and low HALP score was a standalone risk factor for early recurrence and short survival (*Xu et al., 2020*). Similar results were found in another study, which included 355 patients who received radical resection for esophageal squamous cell carcinoma. According to the findings, individuals who had resectable esophageal squamous cell carcinoma had a preoperative HALP score that was a reliable indicator of their prognosis (*Shen et al., 2019*). Clinical biomarkers that are frequently used in daily clinical practice include lymphocytes, albumin, hemoglobin, and platelets. Very few research studies have investigated the association between HALP scores and prognoses in patients with EC. In our study, 22.2 was the optimal cut-off value of HALP score. This is consistent with the findings of some earlier studies, including those on metastatic prostate cancer (32.4 was the cut-off value), pancreatic cancer (44.56 was the cut-off value), small cell lung cancer (25.8 was the cut-off value), bladder cancer (22.2 was the cut-off value), and esophageal cancer (31.8 was the cut-off value) (*Guo et al., 2019*; *Xu et al., 2020*; *Peng et al., 2018*; *Shen et al., 2019*; *Tian et al., 2021*). Numerous studies conducted recently have revealed that a variety of inflammatory and/or nutrition-related indicators, including PNI, PLR, and NLR, are connected to the prognosis of several malignancies, including non-small cell lung cancer (*Mandaliya et al., 2019*; *Sun & Zhang, 2018*; *Jiang et al., 2022*, *2021*). In comparison to other prognostic scores, HALP is regarded as an excellent prognostic indicator. A study on HALP score for prostate cancer patients' prognosis showed that HALP score has higher predictive power than NLR and OLR (*Guo et al., 2019*). Another study on the prognosis of pancreatic cancer also reached the same result (*Xu et al., 2020*).

This study found that HALP score was closely related to differentiation, depth of myometrial invasion, lymph node metastasis and recurrence of EC patients. It implies that it might be connected to the recurrence of EC and lymph node metastases. Further logistic regression analysis demonstrated that the HALP score (OR = 2.216) was the influencing factor for the recurrence, and the HALP score (OR = 2.087) was also the influencing factor for lymph node metastasis. Patients with low HALP scores are more prone to lymph node metastasis and recurrence. Doctors can design more personalized treatment plans based on the results of HALP scores in clinical practice, improve treatment effectiveness, and

improve patient prognosis. ROC curve analysis revealed that the AUC of HALP score in predicting lymph node metastasis and recurrence were 0.871 and 0.855. This indicates that HALP score is a strong diagnostic tool for predicting lymph node metastasis and recurrence in patients. In clinical practice, doctors can use HALP score results for predicting the risk related to lymph node metastasis and recurrence in patients, develop more scientific and personalized treatment plans, and help patients obtain better treatment effect and prognosis. In general, doctors will consider the patient's condition, medical history, laboratory test results and imaging data when formulating personalized treatment strategies. HALP score may be included as one of the indicators to help doctors better assess the metabolic status and nutritional status of patients, and select appropriate treatment options on this basis.

## CONCLUSIONS

HALP score shows good predictive performance in predicting lymph node metastasis and recurrence of EC, which has a profound impact on the medical field, helping to improve the accuracy of prognosis evaluation and treatment effectiveness of endometrial cancer patients, opening up new prospects for the diagnosis and treatment of endometrial cancer, and providing a solid foundation for the medical community to make greater breakthroughs in this field. However, as a retrospective study, all the cases in this study were from our hospital, so the selection bias factor could not be completely ruled out, which needs further study for confirmation. Besides, its specific mechanism of action is still unclear, further research and verification are needed in the future.

### Funding
The study was supported by the National Natural Science Foundation Project (No. 82260313). The funders had no role in study design, data collection and analysis, decision to publish, or preparation of the manuscript.

### Grant Disclosures
The following grant information was disclosed by the authors:
National Natural Science Foundation Project: 82260313.

### Competing Interests
The authors declare that they have no competing interests.

### Author Contributions
- Ying Xiong conceived and designed the experiments, analyzed the data, authored or reviewed drafts of the article, and approved the final draft.
- Yuanyuan Yong conceived and designed the experiments, performed the experiments, prepared figures and/or tables, and approved the final draft.
- Yanhua Wang performed the experiments, analyzed the data, prepared figures and/or tables, and approved the final draft.

## Human Ethics

The following information was supplied relating to ethical approvals (*i.e.*, approving body and any reference numbers):

The medical ethics committee at General Hospital of Ningxia Medical University gave its approval for this investigation.

## Data Availability

The raw data are available in the Supplemental File.

## Supplemental Information

Supplemental information for this article can be found online at http://dx.doi.org/10.7717/peerj.16043#supplemental-information.

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
