# Peer review of "Clinical value of hemoglobin, albumin, lymphocyte, and platelet indexes in predicting lymph node metastasis and recurrence of endometrial cancer: a retrospective study"

_PeerJ, doi:10.7717/peerj.16043_

## Round 0.1 · original submission · Minor Revisions

All three reviewers have identified a number of weaknesses in the study, which require appropriate responses and improvements on your side. I believe that careful refinement of the content of your article, strengthening the credibility of your conclusions by adding the necessary analysis, and improving the language are all needed. Please complete the revision and resubmit the paper as soon as possible.

Reviewer 1 ·

Basic reporting

As a retrospective study, selection bias cannot be completely ruled out. Therefore, further research and verification are needed to understand the specific mechanisms behind this relationship.

Experimental design

Here are a few flaws should be revised:
1. It would be fantastic to include the name of the hospital where the study was conducted in the gynecology department.
2. It would be helpful to provide statistics on the rise in incidence and how much younger the disease is being diagnosed to provide a more concrete sense of the problem.
3. It's imperative to define HALP indexes and their relevance to endometrial cancer. By doing so, readers can better understand the score's significance. Besides, it's not clear whether the HALP score is a newly developed tool or if it's been around for a while. The manuscript should inform readers if the score is a new development and explain its significance in more detail.
4. Please simplify the language used to describe the X-tiles software. This way, it won't be as hard to read or understand.
5. The detailed follow-up process is crucial to the study and deserves emphasis. By showing readers how the study is conducted, it can be better understood and appreciated.
6. You need to clarify the dependent and independent variables studied to improve the text's cohesiveness.
7. I suggest using a flowchart to illustrate the study's methods, including the study design, for better comprehension.
8. Please explain the mechanisms at work for the relationship between the nutritional status of cancer patients, inflammation, and prognosis.
9. You need to parse out more information on how physicians can implement HALP scores in a patient's treatment plan to create more personalized treatment strategies.

Validity of the findings

1. Please address potential criticisms of the study or points of confounding and explain further to build the robustness of the findings. Besides, explain more about the limitations of the study with regards to external validity, population sampling bias, and retrospective nature. Specifically, the number of cases with low differentiation is quite small, so it's unclear whether this subgroup will produce meaningful results. Therefore, the manuscript should address the possibility of spurious findings due to small sample size.
2. Please address potential future research based on the results of this study or the gaps in research which this study has identified.
3. The conclusion is clear, but I suggest rephrasing it to emphasize the significance of the results and the impact on the medical field.

Additional comments

None.

Reviewer 2 ·

Basic reporting

This manuscript examines the predictive value of HALP score for lymph node metastasis and recurrence in EC patients. The article finds that the HALP score can accurately predict these factors and can help develop personalized treatment plans for patients. However, this study has limitations should be amended.

Experimental design

No comment

Validity of the findings

No comment

Additional comments

1) The introduction may benefit from a revision that more wholly encapsulates the study's main objective.
2) It is recommended to include statistics or numbers to demonstrate the significance of the rise in endometrial cancer incidence, underscoring its impact on women's health.
3) To fully contextualize the information, please define the NCCN guidelines' meaning and purpose in this study.
4) The systematic pelvic and para-aortic lymph node resection can be described using simpler language to aid comprehension.
5) Please elaborate on the significance of lymph node metastasis to help readers understand its importance.
6) It is important to clarify why early diagnosis and timely treatment are crucial in endometrial cancer, and what impact it has on the prognosis of patients.
7) Please define what HALP scores are and how they can be used to predict the prognosis of cancer patients, considering the target audience of the paper.
8) The methods of collecting clinical data should be further explained. Specifically, it's unclear how differentiation grade and myometrial invasion were determined.
9) The manuscript should explain why such a follow-up plan was used.
10) More information is needed on how the statistical analysis was conducted. For example, the manuscript should describe what variables were included in the logistic regression model and how the ROC curve was constructed.
11) Please provide a comprehensive and concise explanation of the roles of halp components: hemoglobin, albumin, lymphocyte, and platelet in tumor metabolism, survival, invasion, migration, angiogenesis, and resistance to chemotherapy or radiation therapy.
12) Please emphasize the study's novel contribution to the literature to intrigue readers and convey its significance.
13) Using more specific language to describe the results may facilitate the comprehension of readers.
14) Please define the limitations of the study, particularly with regards to its external validity and population sampling bias.

Reviewer 3 ·

Basic reporting

1. Use more concise language in the introduction to present the study's objective.
2. Provide a clear explanation of the terms used in the study to improve readers' understanding.
3. Define what hematological measures in 1.2.2 are, for clarity and context.
4. Introduce the applied multivariate logistic regression model.
5. The first inclusion criterion listed is unclear. It should specify what kind of pathological examination was used to confirm EC.

Experimental design

1. The fourth exclusion criterion is unclear. The manuscript should specify what level of liver or renal dysfunction would exclude a patient from the study.
2. Consider rephrasing or editing the statistical significance statement in 1.4 to enhance readability.
3. Consider providing more information on the X-tile software to enhance its significance and functionalities.

Validity of the findings

1. Define which methods of univariate and multivariate logistic regression analyses were used, how to apply the univariate and multivariate logistic regression analyses, and the aim of conducting them.
2. The first paragraph of discussion can be more concise and to the point. It can be shortened and rephrased to something like "Malignant cells exhibit traits of proliferation, invasion, and metastasis, and the systemic inflammatory immunological milieu can contribute to these characteristics. The nutritional and metabolic status of the host plays a role as well. The HALP score is a composite measure of hemoglobin, lymphocyte, platelet count, and albumin that reflects both inflammation and nutritional status. We examined its use in predicting lymph node metastasis and recurrence in EC patients in this study."
3. The sentence about HALP being convenient and economical should be removed or rewritten, as it is tangential to the argument and not supported by evidence in results.
4. The fourth paragraph of discussion should provide hazard ratios or confidence intervals for the HALP score as an independent prognostic factor, and specify what other variables were included in the regression model.

Additional comments

1. Provide a brief description of what is displayed in figures and tables to improve data interpretation.

---

## Round 0.2 · accepted · Accept

In carefully evaluating the content of this revised paper, I was satisfied with the responses and revisions made by the authors. The Reviewer's concerns have been well addressed. With the necessary revisions and improvements, the quality of this paper has been significantly improved. I believe that this revised manuscript is ready to be considered for publication in this journal.